# Assessment of Arterial Transit Time and Cerebrovascular Reactivity in Moyamoya Disease by Simultaneous PET/MRI

**DOI:** 10.3390/diagnostics13040756

**Published:** 2023-02-16

**Authors:** Kenji Takata, Hirohiko Kimura, Shota Ishida, Makoto Isozaki, Yoshifumi Higashino, Ken-Ichiro Kikuta, Hidehiko Okazawa, Tetsuya Tsujikawa

**Affiliations:** 1Department of Radiology, Faculty of Medical Sciences, University of Fukui, Fukui 910-1193, Japan; 2Department of Radiological Technology, Faculty of Medical Sciences, Kyoto College of Medical Science, Kyoto 622-0041, Japan; 3Department of Neurosurgery, Faculty of Medical Sciences, University of Fukui, Fukui 910-1193, Japan; 4Biomedical Imaging Research Center, University of Fukui, Fukui 910-1193, Japan

**Keywords:** ASL-CBF, ASL-ATT, PET-CBF, PET-CVR, Moyamoya disease

## Abstract

We investigated the relationship between MRI-arterial spin labeling (ASL) parameters and PET-cerebral blood flow (CBF)/cerebrovascular reactivity (CVR) simultaneously obtained by PET/MRI in Moyamoya disease. Twelve patients underwent ^15^O-water PET/MRI with the acetazolamide (ACZ) challenge test. PET-CBF and PET-CVR were measured using ^15^O-water PET. Pseudo-continuous ASL obtained the robust arterial transit time (ATT) and ASL-CBF estimation. ASL parameters were compared with PET-CBF and PET-CVR. Before ACZ loading, absolute and relative ASL-CBF were significantly correlated with absolute and relative PET-CBF (r = 0.44, *p* < 0.0001, and r = 0.55, *p* < 0.0001, respectively). After ACZ loading, absolute and relative ASL-CBF were significantly correlated with absolute and relative PET-CBF (r = 0.56, *p* < 0.001, and r = 0.75, *p* < 0.0001, respectively), and ΔASL-CBF was significantly correlated with ΔPET-CBF (r = 0.65, *p* < 0.0001). Baseline ASL-ATT had strong negative correlations with ΔPET-CBF and PET-CVR (r = −0.72, *p* < 0.0001, and r = −0.66, *p* < 0.0001, respectively). Baseline ASL-ATT of MCA territories with CVR <30% (1546 ± 79 ms) was significantly higher than that with CVR > 30% (898 ± 197 ms). ASL-ATT ratio of MCA territories with CVR < 30% (94.0 ± 10.5%) was significantly higher than that with CVR > 30% (81.4 ± 11.3%). ATT correction using multiple postlabeling delays increased the accuracy of ASL-CBF quantitation. Baseline ASL-ATT is a hemodynamic parameter and may represent an efficient alternative to PET-CVR.

## 1. Introduction

Positron emission tomography (PET) is a well-established method for hemodynamic evaluation of cerebral blood flow (CBF) and oxygen metabolism such as oxygen extraction fraction (OEF) in steno-occlusive cerebrovascular diseases (CVDs) [1,2]. An elevated OEF determined by PET is believed to represent a critical reduction of cerebral perfusion pressure, also called misery perfusion or stage II ischemia, where CBF reduction and residual oxygen consumption cause a relative elevation of OEF in the brain. The detection of misery perfusion (elevated OEF) is important in assessing the necessities of neurosurgical treatments, including extracranial–intracranial bypass surgery, due to high rates of ischemic events and recurrent strokes. Cerebrovascular reactivity (CVR) is also measured as a risk indicator for symptom recurrence and indication for surgery [3,4]. CVR is defined as the ability to increase CBF in response to vasodilatory stimuli, and a low CVR is associated with higher ischemic stroke risk. The acetazolamide (ACZ) challenge test is widely used to measure CVR in brain PET [5,6].

Arterial spin labeling (ASL) is a magnetic resonance (MR) perfusion assessment method that uses magnetically labeled arterial blood as an endogenous tracer without exposure to radiation [7,8,9]. Recently reported CBF measurements often employ the ASL method [10,11]. Quantitative CBF measurement by ASL requires quantified parameters such as the T1 value of brain tissue, T1 value of arterial blood, arterial blood volume, and, the most important parameter, arterial transit time (ATT) [12]. ATT is the time required for the labeled spins to reach brain tissue from the labeled surface and varies widely with pathology, age, and vascular region. In regions with prolonged ATT, the label effect is smaller due to T1 relaxation of the inverted spin. Therefore, for quantitative CBF evaluation in steno-occlusive CVDs using ASL, it is necessary to account for the label effect reduction associated with ATT prolongation (ATT correction).

Moyamoya disease (MMD) is a chronic steno-occlusive CVD that causes chronic progressive stenosis of internal carotid artery (ICA) endings and abnormal vascular network formation (Moyamoya vessels) in the cerebral basal region as collateral blood flow [13,14], where slow blood flow through collateral blood vessels results in label effect reduction and CBF underestimation due to ATT prolongation in MR-ASL examinations [15,16]. ASL images consisting of multiple postlabeling delays (PLDs) are used to obtain an accurate ATT map [17]. However, acquiring multiple PLD images is difficult in routine clinical examinations due to the long acquisition time [16]. Recently, the Hadamard Multidelay method was devised to improve time efficiency [18]; thus, protocols with clinically applicable scan time were proposed [19].

According to classic basic hemodynamic autoregulation theory, in normal brain perfusion areas with well-preserved CVR, ACZ loading causes peripheral vessels to dilate and vascular resistance to decrease [20]. Decreased vascular resistance is expected to result in faster tissue passage and consequently shorter ATT. In contrast, in areas where the main artery is stenotic or obstructed with collateral pathways, increased vascular resistance is expected to result in slower tissue passage and consequently prolonged ATT. In addition, in the state of misery perfusion, vasodilation by autoregulation has already occurred, and the reactivity to ACZ loading is expected to be reduced.

Based on the above, we hypothesized that ATT correction using multiple PLD images may increase the accuracy of ASL-CBF measurements before and after ACZ loading and that ASL-ATT may be a hemodynamic parameter and correlated with PET-CVR [21,22]. The aim of this study was to assess the association of ASL parameters with ^15^O-water PET-CBF before and after ACZ loading and PET-CVR in MMD simultaneously obtained by integrated PET/MRI.

## 2. Materials and Methods

### 2.1. Patients

This study was approved by the Ethics Committee of University of Fukui, Faculty of Medical Sciences (study protocol # 20170214, 14 February 2017), based on its guidelines (Ethical Guidelines for Medical Science Research with Humans) as well as the Helsinki Declaration of 1975 (revised in 1983). Written informed consent was obtained from each patient. This study prospectively enrolled 12 patients (4 men and 8 women, median age 42 years, age range 11–85 years) with MMD who were undergoing PET/MR examinations with the ACZ loading test performed at our university hospital between May 2017 and February 2020. The stenosis ratio was determined according to the North American Symptomatic Carotid Endarterectomy Trial (NASCET) criteria for cerebral and cervical vessels, a method originally used for evaluating cervical blood vessels (stenosis ratio 50–70%, moderate; 71–100%, severe). Eleven patients were measured based on digital subtraction angiography (DSA), and one patient was measured based on MR angiography (MRA) images. The clinical characteristics of the study population are summarized in Table 1.

### 2.2. MR Imaging

A whole-body PET/MRI scanner (Signa PET/MR, GE Healthcare, Milwaukee, WI, USA) equipped with standard 8-channel head coil was used for simultaneous PET and MRI data acquisition [23]. FSPGR (GE Healthcare) T1-weighted images (repetition time, 8.5 ms; echo time, 3.2 ms; flip angle, 12°; slice thickness, 1.4 mm; 136 slices; pixel size, 1.0 × 1.0 mm) were acquired for structural images.

All ASL imaging was acquired with a pseudocontinuous ASL sequence (pCASL) followed by 3D spiral spin echo readout without vascular crusher gradient. For robust arterial transit time and precise CBF estimation within a clinically acceptable period, we used the combined acquisition of Hadamard-encoded multidelay (He-pCASL) and long-labeled long-delay pseudocontinuous arterial spin labeling (LLLD-pCASL). The details are described in-depth by our previous paper [19], and here in brief. First, three delayed He-pCASLs produces three sub-bolus conditions and their entire bolus, four conditions in total. Second, a single-delay pCASL with a long LD of 4000 ms and long PLD of 3000 ms was also acquired. The exact numerated bolus conditions were as follows: [LD (ms), PLD (ms)] = [958, 700], [1131, 1658], [1909, 2790], [4000, 700], and [4000, 3000] for the first, second, and third sub-boluses of He-pCASL, the LLSD, and the LLLD-pCASL, respectively.

### 2.3. ATT and CBF Calculation on ASL

ATT was calculated using the signal-weighted delay (WD) method from the following Equations (1) and (2) [18] using the ASL signal obtained in each [LD, PLD] condition, which model the ASL signal function being converted to a monotonically increasing function for $\delta_{a}$:(1)$$WD\left(\delta_{a}\right)=\left[\overset{n}{\underset{i=1}{\sum }}w_{i}\cdot \Delta M\left(\delta_{a},w_{i}\right)\right]/\left[\overset{n}{\underset{i=1}{\sum }}\Delta M\left(\delta_{a},w_{i}\right)\right]$$

The single-compartment model was assumed for the perfusion signal in WD function.
(2)$$\begin{array}{l} \frac{\Delta M\left(w_{i}\right)}{M_{0}}=\frac{2\alpha T_{1a}}{\lambda }\cdot f\\ \;\;\;\;\;\;\cdot \;exp\left(-\frac{\delta_{a}}{T_{1a}}\right)\left\{exp\left(\frac{-max\left(w_{i}-\delta_{a},0\right)}{T_{1a}}\right)-exp\left(\frac{-max\left(w_{i}-\delta_{a}+\tau_{i},0\right)}{T_{1a}}\right)\right\} \end{array}$$

In the above equations, ΔM is the perfusion signal, $\delta_{a}$ is the arterial transit time, w*_i_* is the postlabeling delay of the *i*-th [LD, PLD] condition of ASL acquisition, M_0_ is the equilibrium magnetization of arterial blood, α is the labeling efficiency (0.85), *λ* is partition coefficient (0.9), *f* is cerebral blood flow, $T_{1a}$ is the longitudinal relaxation time of arterial blood (1.65 s), and τ is label duration.

Once ATT was determined by the weighted delay method, CBF with ATT correction was calculated. The details how the ATT corrected CBF in the least-squares solution using all [LD, PLD] conditions is previously reported [19]. Figure 1 demonstrates the calculation of ATT corrected ASL-CBF from five perfusion signals, four from Hadamard and the other from LLLD acquisition, in a normal subject.

### 2.4. PET Acquisition and CBF Calculation

Each subject received 555 MBq of ^15^O-water radiotracer before and 15 min after acetazolamide (ACZ) administration. The image-derived input function (IDIF) method was employed for the calculation of PET-CBF without the arterial sampling [24]. After tracer injection, the dynamic 3D PET images were reconstructed from the 3-min list mode TOF data in 21 frames of 12 × 5 s, 6 × 10 s, and 3 × 20 s, and PSF on. Decay in radioactivity of the dynamic PET data was corrected to the starting point of each scan. The dynamic PET data and MRA images were transferred to a workstation (AW4.6, GE Healthcare, Chicago, IL, USA) to obtain an arterial input function from images. Attenuation correction was achieved using the zero-TE MRI-based attenuation correction (ZTE-MRAC) [25].

PET-CBF images were calculated using the following methods: The autoradiography (ARG) method with a partition coefficient of 0.9 [24]. Time-activity curves from the IDIF were used for the arterial input function. The 3-min static PET image was used for the brain tissue count in the ARG method.

### 2.5. Date Analysis

We used the SPM (version 12; The Wellcome Trust Centre for Neuroimaging) for the image registration, standardization, and brain mask creation processes. Final region-of-interest (ROI) value extraction and image-display processing were performed by in-house software written in MATLAB version 2022a (The Math Works, Inc. Natick, MA, USA). First, all images of PET-CBF, ASL-CBF, and ASL-ATT before and after ACZ challenge were co-registered to the same structural whole-brain T1WI. Second, a gray matter (GM) brain mask was created from each subject T1WI by segmentation of the brain using the SPM command. Finally, each subject’s T1WI was converted to a standardized T1-weighted Montreal Neurological Institute (MNI) template in a 152–2mm space along with all CBF, ATT, GM-mask by using the built-in SPM function for rigid-body transformation.

The transit time-based ROI value extraction was used in the comparison between PET and ASL parametric images, which were categorized into 18 regions: the left and right proximal, intermediate, and distal regions of each of the major anterior cerebral artery (ACA), middle cerebral artery (MCA), and the posterior cerebral artery (PCA) territories, which is previously reported and available in MNI standard space [26]. Figure 2 shows transit time-based ROIs. Relative CBF value in each ROI was calculated relative to the whole gray mask CBF value in each PET and ASL-CBF. We selected the MCA ROIs when we compared ROIs with affected and nonaffected cerebral flow based on the PET-CVR less than mean value of 30%.

### 2.6. Statistical Analysis

Regression analyses were performed using Pearson’s correlation coefficient (r) between baseline absolute and relative ASL-CBF and PET-CBF, between absolute and relative ASL-CBF and PET-CBF after ACZ loading, between the changes of ASL-CBF (ΔASL-CBF) and PET-CBF (ΔPET-CBF), and between baseline ASL-ATT and ΔPET-CBF and PET-CVR (percent increase). Regression analyses were performed using all transit time-based ROIs, described in the previous section. The Pearson’s r value is considered to have almost negligible correlation when ranging 0–0.3, low correlation when 0.3–0.5, moderate correlation when 0.5–0.7, and a high correlation when it ranges 0.7–1 [27]. Differences in the ASL-ATTs and ratio between ROIs with and without CVR of MCA territory of less than 30% were assessed using the unpaired t test. All statistical analyses were performed using SPSS statistics version 22. *p* < 0.05 was considered significant, except that *p* < 0.05/3 = 0.017 (Bonferroni correction) was considered significant between baseline ASL-ATT and ΔPET-CBF and PET-CVR.

## 3. Results

### 3.1. Correlation of ASL-CBF with PET-CBF

Before ACZ loading (baseline), absolute and relative ASL-CBF were significantly correlated with absolute and relative PET-CBF, respectively (Pearson’s r = 0.44, *p* < 0.0001, and r = 0.55, *p* < 0.0001, respectively) (Figure 3a,b). After ACZ loading, absolute and relative ASL-CBF were significantly correlated with absolute and relative PET-CBF, respectively (r = 0.56, *p* < 0.001, and r = 0.75, *p* < 0.0001, respectively) (Figure 3c,d), and ΔASL-CBF was significantly correlated with ΔPET-CBF (r = 0.65, *p* < 0.0001) (Figure 3e).

### 3.2. Correlation and Comparison of Baseline ASL-ATT with PET-CBF Changes

Baseline ASL-ATT had strong negative correlations with ΔPET-CBF and PET-CVR (r = −0.72, *p* < 0.0001, and r = −0.66, *p* < 0.0001, respectively) (Figure 4a,b). Baseline ASL-ATT of MCA territories with PET-CVR less than 30% (mean ± SD = 1546 ± 379 ms) was significantly higher than that with PET-CVR more than 30% (898 ± 197 ms) (Figure 4c). ASL-ATT ratio of MCA territories with PET-CVR less than 30% (94.0 ± 10.5%) was significantly higher than that with PET-CVR more than 30% (81.4 ± 11.3%) (Figure 4d).

### 3.3. Representative Images

Figure 5 shows representative images of a 46-year-old woman with severe MMD (patient No. 6) with transient ischemic attack (TIA) symptoms. A T2-weighted MR image (a) shows no abnormal findings in the brain parenchyma; however, MR angiography (b) demonstrates severe stenosis of the right ICA. Despite the slight overestimation of ASL-CBF compared with PET-CBF, baseline and post-ACZ ASL-CBF with ATT correction (c,d) are similar to baseline and post-ACZ PET-CBF (e,f), respectively. Baseline ASL-ATT was prolonged in the right MCA territory (g). Although the prolonged ASL-ATT in the right MCA territory remained about the same even after ACZ loading, post-ACZ ASL-ATT was shortened in the left MCA territory due to the decreased vascular resistance (h). Decreases in ΔPET-CBF and PET-CVR in the right MCA territory (i,j, respectively) were similar to a negative/positive inversion of prolonged ASL-ATT.

## 4. Discussion

The present study demonstrated two benefits of ASL-ATT measurements for evaluating ASL-CBF in MMD patients with chronic severe stenosis or occlusion of ICA endings and formation of Moyamoya vessels. Firstly, ATT correction using multiple PLD images compensated for the label effect reduction and underestimation of ASL-CBF and increased the accuracy of ASL-CBF quantitation both before and after ACZ loading. Secondly, baseline ASL-ATT was significantly and negatively correlated with PET-CVR measured using the ACZ challenge and may present an efficient alternative.

CVR is important in chronic steno-occlusive CVDs for assessing risks of symptom recurrence and surgery indications. Patients with reduced CVR reportedly have a significantly higher risk of recurrent stroke compared to patients with normal CVR [3,4]. However, previous studies reported that when a threshold determined by only CVR was applied for misery perfusion detection, approximately half of the patients with reduced CVR were identified as false positives [28,29,30]. Okazawa et al. demonstrated that when the reduction in baseline CBF and CVR were combined, specificity of detection of misery perfusion improved as compared with using CVR alone [31].

Although an ASL-ATT map calculated using multiple PLD images was originally used for correcting ASL-CBF values, the present results suggested that ASL-ATT may also be used as a hemodynamic parameter, i.e., a substitute for PET-CVR [32]. Areas where major vessels are stenotic/obstructed have already undergone vasodilation by compensatory mechanisms resulting in decreased PET-CVR. In these areas, collateral vessels such as Moyamoya vessels develop but are too weak to maintain native perfusion and can be causes of prolonged ASL-ATT. In contrast, in normal areas, PET-CVR is preserved, and ACZ loading leads to a decrease in vascular resistance due to vasodilation, resulting in ASL-ATT shortening. The combined use of baseline ASL-CBF and ASL-ATT may be comparable to PET-CBF and PET-CVR measurements in detecting misery perfusion and predicting the risk of symptom recurrence in MMD patients.

ASL-CBF measurements combined with ASL-ATT have comparative advantages over PET-CBF and PET-CVR measurements. The use of baseline ASL parameters without ACZ loading is less invasive and more patient-friendly, requiring only ASL images consisting of multiple PLDs compared to using ^15^O-water PET with ACZ challenge to obtain baseline and post-ACZ PET-CBF and PET-CVR. Baseline ASL-CBF and ASL-ATT measurements do not require the PET cyclotron and radiopharmacy facility and can avoid radiation exposure and the adverse effects of intravenous ACZ administration. The adverse effects include headache, nausea, dizziness, tinnitus, numbness of the extremities, motor weakness of the extremities, and general malaise, especially in young females [33]. Baseline ASL-CBF and ASL-ATT measurements are safe and easy to install to evaluate misery perfusion areas in Moyamoya disease patients.

However, ASL-CBF measurements have a comparative disadvantage over PET-CBF measurements. In the present study, ASL-CBF tended to be overestimated compared to PET-CBF in hypo- and normal-perfusion areas (Figure 3). Previous studies reported that the ATT was prolonged in the affected areas of acute infarction or chronic steno-occlusive CVDs and that labeled spins were retained more in the vessels of the affected areas than in healthy areas [34,35]. This phenomenon was identified as arterial transit delay. Similar arterial transit artifacts have been reported in MMD, which in turn are reported as clinically useful signs for evaluating the presence and intensity of collateral flow [36]. In any event, Moyamoya vessels can be the sources of residual labeled spins in MMD leading to CBF measurement errors (overestimation). In addition, ACZ loading causes dilation of microvessels and increases cerebral blood volume resulting in an increase in intravascular signals, especially in the normal-perfusion areas. The measurement errors can be reduced by incorporating protocols to suppress signals in the vessels [37]. However, in the present study, we did not suppress intravascular signals to investigate the hemodynamic changes both in the affected and healthy areas as the intravascular signal suppression may offset changes in ASL-CBF before and after ACZ loading.

Study limitations include the small number of participants and the large variation of absolute ASL-CBF. Although the results demonstrate the significant correlations between ASL-CBF with ATT correction and PET-CBF and between ASL-ATT and PET-CVR in MMD patients, this method remains impractical for immediate clinical practice. Further studies with more MMD patients are necessary to confirm our preliminary results. The correlation between absolute ASL-CBF and PET-CBF was improved after ACZ loading (r = 0.56, *p* < 0.001), likely due to ^15^O-water PET reflecting intravascular tracers, which is similar to ASL-CBF measurements affected by intravascular signals; thus, the ACZ challenge may enhance the correlation between ASL-CBF and PET-CBF. Therefore, improved ASL methods for suppressing residual intravascular signals while maintaining the quantitative performance of ASL-CBF and ASL-ATT are still required [38].

## 5. Conclusions

In MMD, ATT correction using multiple PLD images compensated for the label effect reduction and underestimation of ASL-CBF and increased the accuracy of ASL-CBF quantitation both before and after ACZ loading. Baseline ASL-ATT was significantly and negatively correlated with PET-CVR measured using the ACZ challenge and may represent an efficient alternative to it.

## Figures and Tables

**Figure 1 diagnostics-13-00756-f001:**
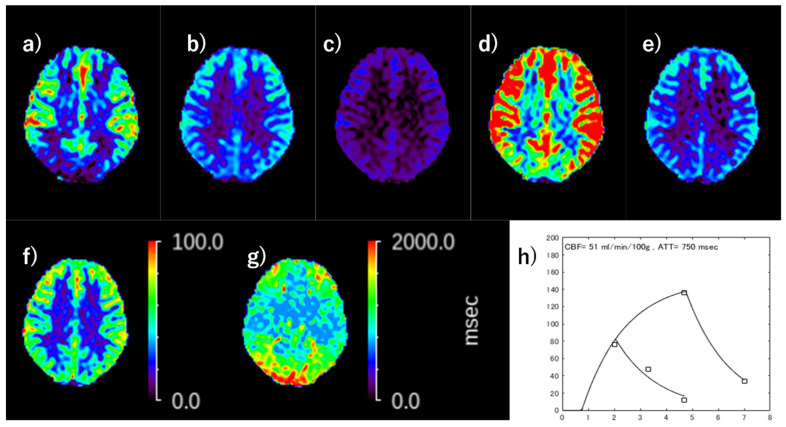
The calculation of ATT corrected ASL-CBF using the combined Hadamard and long-labeled long-delay (LLLD) acquisitions. (**a**) Perfusion signal (Δ*M*/*M*_0_) map of [LD, PLD] = [1.33, 0.7], (**b**) Δ*M*/*M*_0_ map of [LD, PLD] = [1.33, 2.03], (**c**) Δ*M*/*M*_0_ map of [LD, PLD] = [1.33, 3.36], (**d**) Δ*M*/*M*_0_ map of [LD, PLD] = [4.0, 0.7], (**e**) Δ*M*/*M*_0_ map of [LD, PLD] = [4.0, 3.0], (**f**) ATT-corrected CBF, (**g**) ATT map, (**h**) plots of each Δ*M*/*M*_0_ and the draw of selected least square solution. Plotted points are in left cerebral cortical pixel in the central semiovale section of the maps of (**a**–**e**), respectively.

**Figure 2 diagnostics-13-00756-f002:**
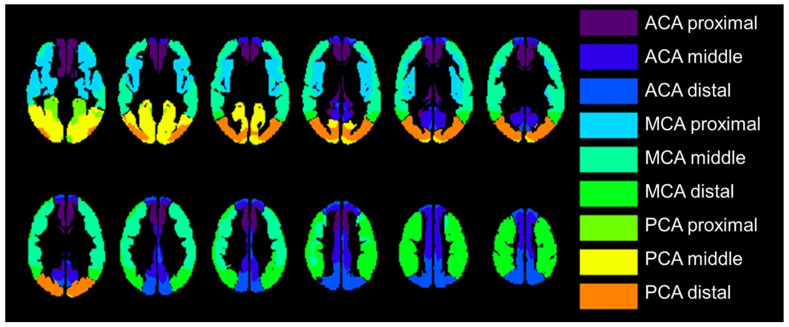
The regions of interest (ROI) were placed proximal, middle, and distal to the major vascular territories with structures located in the deep gray matter not included (e.g., basal ganglia and thalamus); 18 regions were measured per case. ACA, anterior cerebral artery; MCA, middle cerebral artery; PCA, posterior cerebral artery.

**Figure 3 diagnostics-13-00756-f003:**
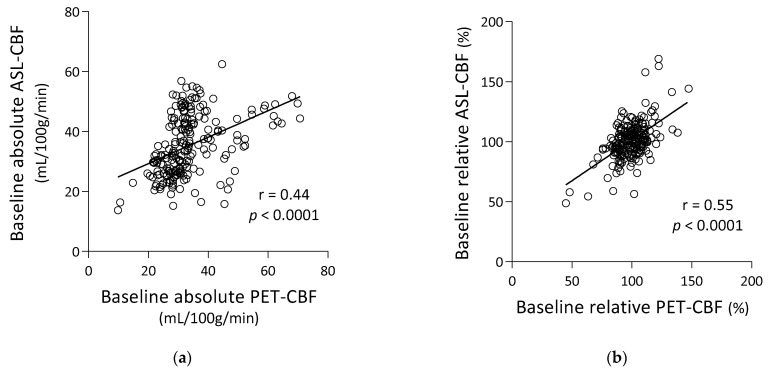
Correlation between PET-CBF and ASL-CBF. (**a**) Pre-ACZ absolute PET-CBF and ASL-CBF, (**b**) pre-ACZ relative PET-CBF and ASL-CBF, (**c**) post-ACZ absolute PET-CBF and ASL-CBF, (**d**) post-ACZ relative PET-CBF and ASL-CBF, (**e**) differences in ASL-CBF (ΔASL-CBF) and PET-CBF (ΔPET-CBF) between baseline and post-ACZ states. Regression lines are shown with Pearson’s correlation coefficients (r) and associated *p*-values.

**Figure 4 diagnostics-13-00756-f004:**
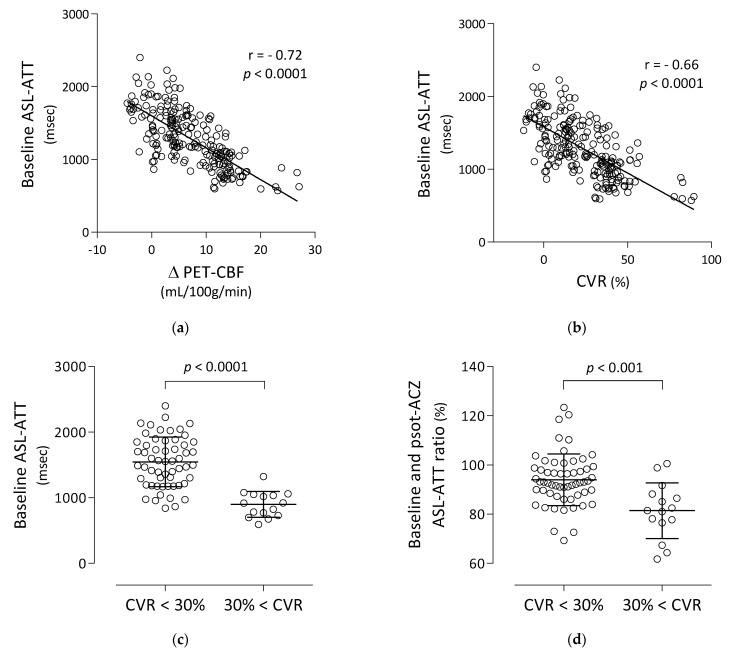
Correlation between pre-ACZ ASL-ATT and ΔPET-CBF (**a**) and CVR (**b**). Regression lines are shown with Pearson’s correlation coefficients (r) and associated *p*-values. Group comparisons of ASL-ATTs (**c**) and the ratio (**d**) between ROIs with and without CVR of MCA territory less than 30%. The significance of differences was assessed by unpaired t test.

**Figure 5 diagnostics-13-00756-f005:**
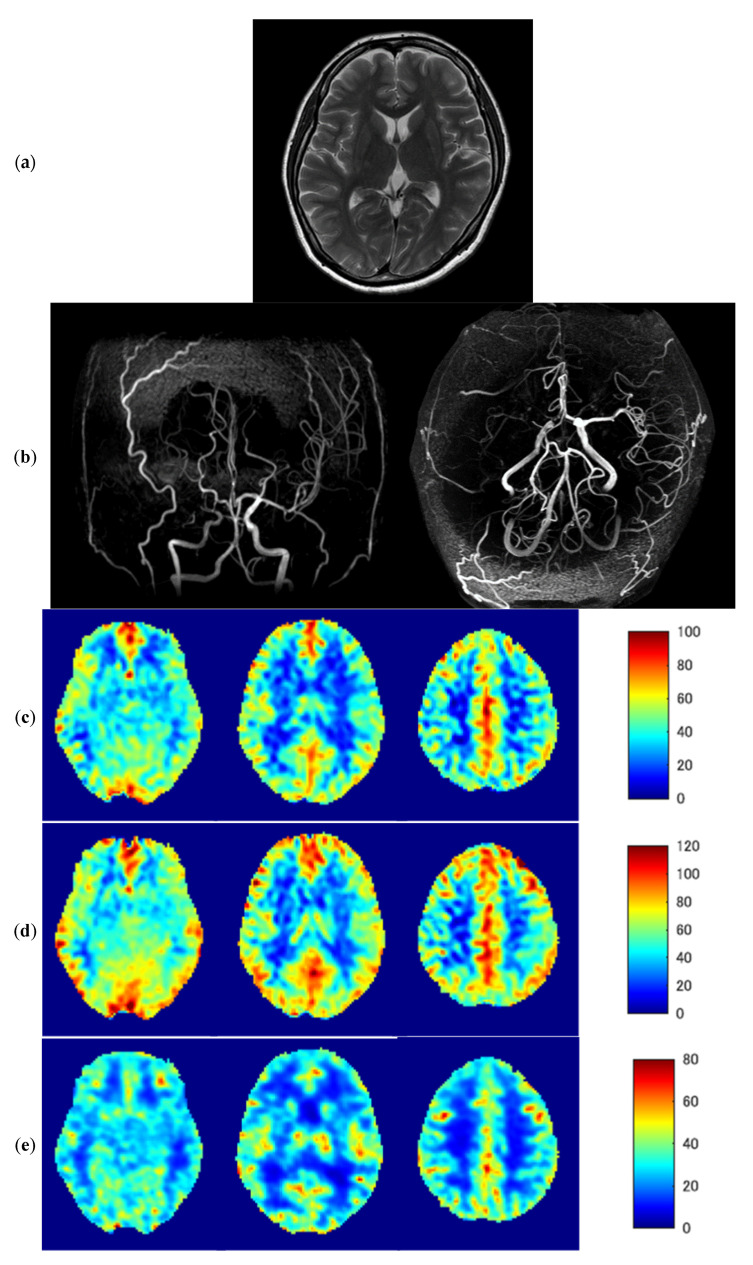
Representative images of a 46-year-old woman with severe Moyamoya disease (patient No. 6). T2-weighted MR image (**a**), MR angiography (**b**), baseline ASL-CBF (**c**), post-ACZ ASL-CBF (**d**), baseline PET-CBF (**e**), post-ACZ PET/CBF (**f**), baseline ASL-ATT (**g**), post-ACZ ASL-ATT (**h**), ΔPET-CBF (**i**), and CVR (**j**) CVR; cerebrovascular reactivity.

**Table 1 diagnostics-13-00756-t001:** Patient characteristics and clinical data.

Patient	Age	Sex	Stenosis/Occlusion	Severity	Symptom
1	44	F	R MCA	severe	asymptomatic
2	85	M	L ICA	severe	hemiparesis
3	44	F	R ICA, MCA, A1. L ICA top	severe	hemiparesis
4	38	F	R ICA top, MCA, A1	severe	hemiparesis
5	51	M	R ICA, MCA. R/L ACA	severe	hemiparesis, dysarthria
6	46	F	R ICA top, MCA, A1	severe	TIA
7	46	M	R ICA top, MCA, A1	severe	cerebral hemorrhage
8	45	F	R/L ICA top, MCA	severe	higher brain dysfunction
9	42	F	R/L ICA top	severe	TIA
10	42	M	R ICA top, MCA	severe	asymptomatic
11	15	F	R/L ICA top, MCA	severe	headache
12	11	F	R/L ICA top, MCA	severe	headache

The stenosis ratio was determined according to the North American Symptomatic Carotid Endarterectomy Trial (NASCET) criteria for cerebral and cervical vessels, a method originally used for evaluating cervical blood vessels (stenosis ratio 50–70%, moderate; 71–100%, severe). Note: male (M); female (F); right (R); left (L); middle cerebral artery (MCA); anterior cerebral artery (ACA); internal carotid artery (ICA); first segment of anterior cerebral artery (A1); transient ischemic attack (TIA).

## Data Availability

The files/data used to support the findings of this study are available from the corresponding author upon request.

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
