# Peer review of "Assessment of Arterial Transit Time and Cerebrovascular Reactivity in Moyamoya Disease by Simultaneous PET/MRI"

_diagnostics, 2023, doi:10.3390/diagnostics13040756_

Round 1

Reviewer 1 Report

The authors have assessed the Arterial Transit Time and Cerebrovascular Reactivity in Moyamoya Disease by Simultaneous PET/MRI. The work is well presented and has useful implications in the evaluation of Moyamoya Angiopathy patients.

There are a few typos in the manuscript that require correction.

Author Response

Reply to Reviewer 1

  1. There are a few typos in the manuscript that require correction.

   According to the comment, the typo was corrected. (Line 280)

Reviewer 2 Report

Thank you for the opportunity to review this manuscript. The authors have conducted innovative work by conducting comparative analysis between MRI and PET in terms of arterial transit time and cerebrovascular reactivity in Moya-Moya disease patients. Overall, this is a well written manuscript with sound statistical analysis. However, there is one major limitation in the analysis. Given the small sample size of the study population, I would recommend that authors make corrections for multiple comparisons in their analyses to address the chance of type I error. Looking at the p-values for their analysis, it is unlikely to alter any results, but it is still the prudent approach for scientific publications. Secondly, if authors are hypothesizing that ASL may be a substitute for acetazolamide challenge test, I would recommend authors more adeptly delineate the comparative advantages and disadvantages of the two techniques which can assist the readers to discern the utility of ASL as an alternative approach and design future studies with appropriate designs.

Author Response

Reply to Reviewer 2

  1. Given the small sample size of the study population, I would recommend that authors make corrections for multiple comparisons in their analyses to address the chance of type I error. Looking at the p-values for their analysis, it is unlikely to alter any results, but it is still the prudent approach for scientific publications.

According to the comments, we explained that p < 0.05/3 = 0.017 (Bonferroni correction) was considered significant between baseline ASL-ATT and ΔPET-CBF and PET-CVR. (Line 193 - 195)

  1. Secondly, if authors are hypothesizing that ASL may be a substitute for acetazolamide challenge test, I would recommend authors more adeptly delineate the comparative advantages and disadvantages of the two techniques which can assist the readers to discern the utility of ASL as an alternative approach and design future studies with appropriate designs.

   According to the comments, the statements about comparative advantages and disadvantages were added in Discussion. (Line 264 – 290)

Round 2

Reviewer 2 Report

Thank you for the opportunities to re-review this manuscript. The authors have appropriately addressed my comments.